# ACTIVE FEATURE ACQUISITION WITH GENERATIVE SURROGATE MODELS

## ABSTRACT

Many real-world situations allow for the acquisition of additional relevant information when making an assessment with limited or uncertain data. However, traditional ML approaches either require all features to be acquired beforehand or regard part of them as missing data that cannot be acquired. In this work, we propose models that perform active feature acquisition (AFA) to improve the prediction assessments at evaluation time. We formulate the AFA problem as a Markov decision process (MDP) and resolve it using reinforcement learning (RL). The AFA problem yields sparse rewards and contains a high-dimensional complicated action space. Thus, we propose learning a generative surrogate model that captures the complicated dependencies among input features to assess potential information gain from acquisitions. We also leverage the generative surrogate model to provide intermediate rewards and auxiliary information to the agent. Furthermore, we extend AFA in a task we coin *active instance recognition* (AIR) for the unsupervised case where the target variables are the unobserved features themselves and the goal is to collect information for a particular instance in a cost-efficient way. Empirical results demonstrate that our approach achieves considerably better performance than previous state of the art methods on both supervised and unsupervised tasks.

## 1 INTRODUCTION

A typical machine learning paradigm for discriminative tasks is to learn the distribution of an output, $y$ given a complete set of features, $x \in \mathbb{R}^d$: $p(y \mid x)$. Although this paradigm is successful in a multitude of domains, it is incongruous with the expectations of many real-world intelligent systems in two key ways: first, it assumes that a complete set of features has been observed; second, as a consequence, it also assumes that no additional information (features) of an instance may be obtained at evaluation time. These assumptions often do not hold; human agents routinely reason over instances with incomplete data and decide when and what additional information to obtain. For example, consider a doctor diagnosing a patient. The doctor usually has not observed all possible measurements (such as blood samples, x-rays, etc.) for the patient. He/she is not forced to make a diagnosis based on the observed measurements; instead, he/she may dynamically decide to take more measurements to help determine the diagnosis. Of course, the next measurement to make (feature to observe), if any, will depend on the values of the already observed features; thus, the doctor may determine a different set of features to observe from patient to patient (instance to instance) depending on the values of the features that were observed. Hence, not each patient will have the same subset of features selected (as would be the case with typical feature selection). Furthermore, acquiring features typically involves some cost (in time, money and risk), and intelligent systems are expected to automatically balance the cost and the return on improvement of the task performance. In order to more closely match the needs of many real-world applications, we propose an active feature acquisition (AFA) model that not only makes predictions with incomplete/missing features, but also determines what next feature would be the most valuable to obtain for a particular instance.

In this work, we formulate the active feature acquisition problem as a Markov decision process (MDP), where the state is the set of currently observed features and the action is the next feature to acquire. We also introduce a special action to indicate whether to stop the acquisition process and make a final prediction. Reinforcement learning is then utilized to optimize the MDP, and the agent learns a policy for selecting which next feature to acquire based on the current state.

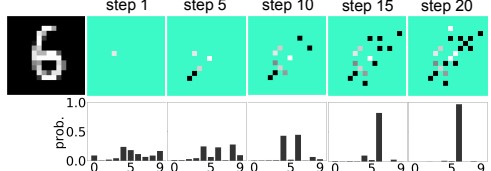

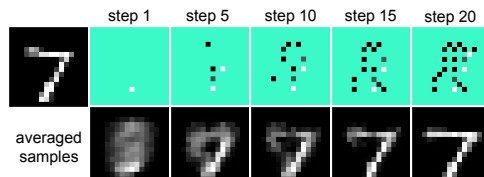

Figure 1: Active feature acquisition on MNIST. Example of the acquisition process and the corresponding prediction probabilities.

Figure 2: Active instance recognition on MNIST. Example of the acquisition process and the averaged inpaintings.

After acquiring its value and paying the acquisition cost, the newly acquired feature is added to the observed subset and the agent proceeds to the next acquisition step. Once the agent decides to terminate the acquisition, it makes a final prediction based on the features acquired thus far. For example, in an image classification task (Fig. 1), the agent would dynamically acquire pixels until it is certain of the image class. The goal of the agent is to maximize the prediction performance while minimizing the acquisition cost.

In the aforementioned MDP, the agent pays the acquisition cost at each acquisition step but only receives a reward about the prediction after completing the acquisition process. To reduce the sparsity of the rewards and simplify the credit assignment problem for potentially long episodes (Minsky, 1961; Sutton, 1988), we leverage a surrogate model to provide intermediate rewards. The surrogate model captures the arbitrary conditional distribution $p(y, x_u \mid x_o)$, where $y$ is the target variable and $u, o \subseteq \{1, \ldots, d\}$ are arbitrary subsets of all $d$-dimensional features. Note that the surrogate model must be able to capture arbitrary conditionals (for subsets $u, o$) since the acquired features will vary from instance to instance. We propose using the surrogate model to calculate intermediate rewards by assessing the information gain of the newly acquired feature, which quantifies how much our confidence about the prediction improves by acquiring this feature.

In addition to producing intermediate rewards, we also propose using the surrogate model to provide side information that assists the agent. First, in order to inform the agent of the current information held in observed features, we pass uncertainty on the target through $p(y \mid x_o)$. Second, to inform the agent about potential values for unobserved features, we pass imputed values by sampling $\hat{x}_u \sim p(x_u \mid x_o)$. Lastly, to inform the agent about the expected utility of acquisitions, we pass an estimate of the expected information gain of acquisitions $i$ for the target variable, i.e., $H(y \mid x_o) - \mathbb{E}_{p(x_i \mid x_o)} H(y \mid x_i, x_o)$. We note that the expected information gain can be used to directly build a greedy policy, where the next feature to acquire is the one maximizes the expected information gain (Ma et al., 2018; Gong et al., 2019). In contrast, our agent learns a non-greedy policy to maximize the long-term returns and use the greedy approach as a 'prior' policy to guide our agent.

In summary, our agent actively acquires new feature and pays the acquisition cost until it decides to terminate the acquisition process and make a final prediction. Meanwhile, the surrogate model calculates the information gain of the acquired feature as an intermediate reward and provides side information to assist the agent in assessing its current uncertainty and help it 'look ahead' to expected outcomes from future acquisitions. When the acquisition process is completed, the environment provides a final reward based on the agent's prediction. Note that the environment does have access to the ground-truth target $y$ to evaluate the reward, but cannot reveal it to the agent. Equipped with the surrogate model, our method, denoted as *GSMRL*, essentially combines model-free and model-based RL into a holistic framework.

Above we discussed AFA for supervised tasks, where the goal is to acquire new features to predict a target variable $y$. In some cases, however, there may not be a single target variable, but instead the target of interest may be the remaining unobserved features themselves. That is, rather than reduce the uncertainty with respect to some desired output response (that cannot be directly queried and must be predicted), we now propose *active instance recognition* (AIR), where the task is to query as few features as possible that allows the agent to correctly uncover the remaining unobserved features. For example, in image data AIR, an agent queries new pixels until it can reliably uncover the remaining pixels (see Fig. 2). AIR is especially relevant in survey tasks, which are broadly applicable across various domains and applications. Most surveys aim to discover a broad set of underlying characteristics of instances (e.g., citizens in a census) using a limited number of queries (questions in the census form), which is at the core of AIR. Policies for AIR would build a personalized subset

---

**Algorithm 1:** Active Feature Acquisition with GSMRL

---

1. load pretrained surrogate model $M$, agent *agent* and prediction model $f_\theta(\cdot)$;
2. instantiate an environment with data $D$ and surrogate model $M$: *env* = Environment$(D, M)$;
$x_o$, o, done, reward = *env*.reset(); // $o = \emptyset$, done=False, reward=0
**while** *not done* **do**
    aux = $M$.query$(x_o, o)$; // query M for auxiliary information
    // aux contains the prediction $\hat{y} \sim p(y \mid x_o)$ and output likelihoods,
    // the imputed values $\hat{x}_u \sim p(x_u \mid x_o)$ and their uncertainties,
    // and estimated utilities $\mathcal{U}_i$ for each $i \in u$ (equation 4).
    action = *agent*.act$(x_o, o, aux)$; // act based on the state and auxiliary info
    $x_o$, o, done, r = *env*.step(action); // take a step based on the action
    // if action indicates termination: done=True, r=$-\mathcal{L}(\hat{y}(x_o), y)$
    // else: done=False, r=$r_m(s, action) - \alpha\mathcal{C}(action)$, $o = o \cup action$
    reward = reward + r; // accumulate rewards
**end**
prediction = $M$.predict$(x_o, o, aux)$ or $f_\theta(x_o, o, aux)$; // make a final prediction

---

of survey questions (for individual instances) that quickly uncovered the likely answers to all remaining questions. To adapt our GSMRL framework to AIR, we set the target variable $y$ equal to $x$ and modify the surrogate model accordingly.

Our contributions are as follows: 1) We propose a way of building surrogate models for AFA problem that captures the state transitions with arbitrary conditional distributions. 2) We leverage the surrogate model to provide intermediate rewards as training signals and to provide auxiliary information that assists the agent. Our framework represents a novel combination of model-free and model-based RL. 3) We extend the active feature acquisition problem to an unsupervised case where the target variables are the unobserved features themselves. Our RL agent can be adapted to this problem with simple modifications. 4) We achieve state-of-the-art performance on both supervised and unsupervised tasks. 5) We open-source a standardized environment inheriting the OpenAI gym interfaces (Brockman et al., 2016) to assist future research on active feature acquisition. Code will be released upon publication.

## 2 METHODS

In this section, we first describe our GSMRL framework for both active feature acquisition (AFA) and active instance recognition (AIR) problems. We then develop our RL algorithm and the corresponding surrogate models for different settings. We also introduce a special application that acquires features for time series data.

### 2.1 AFA AND AIR WITH GSMRL

Consider a discriminative task with features $x \in \mathbb{R}^d$ and target $y$. Instead of predicting the target by first collecting all the features, we perform a sequential feature acquisition process in which we start from an empty set of features and actively acquire more features. There is typically a cost associated with features and the goal is to maximize the task performance while minimizing the acquisition cost, i.e.,

$$\text{minimize } \mathcal{L}(\hat{y}(x_o), y) + \alpha\mathcal{C}(o), \tag{1}$$

where $\mathcal{L}(\hat{y}(x_o), y)$ represents the loss function between the prediction $\hat{y}(x_o)$ and the target $y$. Note that the prediction is made with the acquired feature subset $x_o, o \subseteq \{1, \ldots, d\}$. Therefore the agent should be able to predict with arbitrary subset as inputs. $\mathcal{C}(o)$ represents the acquisition cost of the acquired features $o$. The hyperparameter $\alpha$ controls the trade-off between prediction loss and acquisition cost. For unsupervised tasks, the target variable $y$ is equal to $x$; that is, we acquire features actively to represent the instance with a selected subset.

In order to solve the optimization problem in equation 1, we formulate it as a Markov decision process as done in (Shim et al., 2018):

$$s = [o, x_o], \quad a \in u \cup \phi, \quad r(s, a) = -\mathcal{L}(\hat{y}, y)\mathbb{I}(a = \phi) - \alpha\mathcal{C}(a)\mathbb{I}(a \neq \phi). \tag{2}$$

The state $s$ is the current acquired feature subset $o \subseteq \{1, \ldots, d\}$ and their values $x_o$. The action space contains the remaining candidate features $u = \{1, \ldots, d\} \setminus o$ and a special action $\phi$ that in-

dicates the termination of the acquisition process. To optimize the MDP, a reinforcement learning agent acts based on the observed state and receives rewards from the environment. When the agent acquires a new feature $i$, the current state transits to a new state following $o \xrightarrow{i} o \cup i, x_o \xrightarrow{i} x_o \cup x_i$, and the reward is the negative acquisition cost of this feature. Note $x_i$ is obtained from the environment (i.e. we observe the true $i_{th}$ feature value for the instance). When the agent terminates the acquisition and makes a prediction, the reward equals to the negative prediction loss using current acquired features. Since the prediction is made at the end of the acquisition, the reward of the prediction is received only when the agent decide to terminate the acquisition process. This is a typical temporal credit assignment problem for RL algorithms, which could affect the learning of the agent (Minsky, 1961; Sutton, 1988). In order to remedy this issue, we propose to leverage a generative surrogate model to provide intermediate rewards for each acquisition. The surrogate model estimates the state transitions with arbitrary conditional distributions $p(y, x_u \mid x_o)$ for arbitrary subsets $u$ and $o$. We propose using the surrogate model to assess the intermediate reward $r_m$ for a newly acquired feature $i$. The intermediate rewards are inspired by the information gain to the target variable

$$r_m(s, i) = H(y \mid x_o) - \gamma H(y \mid x_o, x_i), \tag{3}$$

where $\gamma$ is a discount factor for the MDP. In appendix A, we show that our intermediate rewards will not change the optimal policy.

In addition to intermediate rewards, we propose using the surrogate model to also provide side information to assist the agent, which includes the current prediction and output likelihood, the possible values and corresponding uncertainties of the unobserved features, and the estimated utilities of the candidate acquisitions. The current prediction $\hat{y}$ and likelihood $p(y \mid x_o)$ inform the agent about its confidence, which can help the agent determine whether to stop the acquisition. The imputed values and uncertainties of the unobserved features give the agent the ability to look ahead into and future and guide its exploration. For example, if the surrogate model is very confident about the value of a currently unobserved feature, then acquiring it would be redundant. The utility of a feature $i$ is estimated by its expected information gain to the target variable:

$$\mathcal{U}_i = H(y \mid x_o) - \mathbb{E}_{p(x_i \mid x_o)} H(y \mid x_i, x_o) = H(x_i \mid x_o) - \mathbb{E}_{p(y \mid x_o)} H(x_i \mid y, x_o), \tag{4}$$

where the surrogate model is used to estimate the entropies. The utility essentially quantifies the conditional mutual information $I(x_i; y \mid x_o)$ between each candidate feature and the target variable. A greedy policy can be easily built based on the utilities where the next feature to acquire is the one with maximum utility (Ma et al., 2018; Gong et al., 2019). Here, our agent takes the utilities as side information to help balance exploration and exploitation, and eventually learns a non-greedy policy.

When the agent deems that acquisition is complete, it makes a final prediction based on the acquired feaures thus far. The final prediction may be made using the surrogate model, i.e., $p(y \mid x_o)$, but it might be beneficial to train predictions specifically based on the agent's own distribution of acquired features $o$, since the surrogate model is agnostic to the feature acquisition policy of the agent. Therefore, we optionally build a prediction model $f_\theta(\cdot)$ that takes both the current state $x_o$ and the side information as inputs (i.e. the same inputs as the policy). The prediction model can be trained simultaneously with the policy as an auxiliary task; weight sharing between the policy and prediction function helps facilitate the learning of more meaningful representations. Now we have two predictions, from the surrogate model and the prediction model respectively. The final reward $-\mathcal{L}(\hat{y}, y)$ during training is the maximum one using either predictions. During test time, we choose one prediction based on validation performance. An illustration of our framework is presented in Fig. 3. Please refer to Algorithm 1 for the pseudo-code of the acquisition process with our GSMRL framework. We will expound on the surrogate models for different settings below.

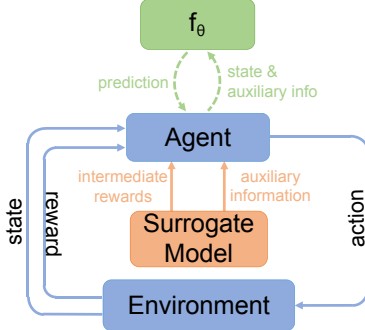

Figure 3: Illustration of our GSMRL framework with an optional prediction model. Dashed arrows connect optional prediction model $f_\theta$.

### 2.1.1 SURROGATE MODEL FOR AFA

As we mentioned above, the surrogate model learns the conditional distributions $p(y, x_u \mid x_o)$. Note that both $x_u$ and $x_o$ are arbitrary subset of the features since the surrogate model must be able

to assist arbitrary policies. Thus, there are $d!$ different conditionals that the surrogate model must estimate for a $d$-dimensional feature space. Therefore, learning a separate model for each different conditional is intractable. Fortunately, Ivanov et al. (2018) and Li et al. (2019) have proposed models to learn arbitrary conditional distributions $p(x_u \mid x_o)$. They regard different conditionals as different tasks and train VAE and normalizing flow based generative models, respectively, in a multi-task fashion to capture the arbitrary conditionals with a unified model. In this work, we leverage the ACFlow model (Li et al., 2019) and extend it to model the target variable $y$ as well. For continuous target variables, we concatenate them with the features, thus $p(y, x_u \mid x_o)$ can be directly modeled with ACFlow. For discrete target variables, we use Bayes' rule

$$p(y, x_u \mid x_o) = \frac{p(x_u \mid y, x_o)p(x_o \mid y)P(y)}{\sum_{y'} p(x_o \mid y')P(y')}. \tag{5}$$

We employ a variant of the ACFlow model that conditions on the target $y$ to obtain the arbitrary conditional likelihoods $p(x_u \mid y, x_o)$ and $p(x_o \mid y)$ in equation 5.

Given a trained surrogate model, the prediction $p(y \mid x_o)$, the information gain in equation 3, and the utilities in equation 4 can all be estimated using the arbitrary conditionals. For continuous target variables, the prediction can be estimated by drawing samples from $p(y \mid x_o)$, and we express their uncertainties using sample variances. We calculate the entropy terms in equation 3 with Monte Carlo estimations. The utility in equation 4 can be further simplified as

$$\mathcal{U}_i = \mathbb{E}_{p(y,x_i|x_o)} \log \frac{p(x_i, y \mid x_o)}{p(y \mid x_o)p(x_i \mid x_o)} = \mathbb{E}_{p(y,x_i|x_o)} \log \frac{p(y \mid x_i, x_o)}{p(y \mid x_o)}. \tag{6}$$

We then perform a Monte Carlo estimation by sampling from $p(x_i, y \mid x_o)$. Note that $p(y \mid x_i, x_o)$ is evaluated on sampled $x_i$ rather than the exact value, since we have not acquired its value yet.

For discrete target variables, we employ Bayes' rule to make a prediction

$$P(y \mid x_o) = \frac{p(x_o \mid y)P(y)}{\sum_{y'} p(x_o \mid y')P(y')} = \mathrm{softmax}_y(\log p(x_o \mid y') + \log P(y')), \tag{7}$$

and the uncertainty is expressed as the prediction probability. The information gain in equation 3 can be estimated analytically, since the entropy for a categorical distribution is analytically available. To estimate the utility, we further simplify equation 6 to

$$\mathcal{U}_i = \mathbb{E}_{p(x_i|x_o)P(y|x_i,x_o)} \log \frac{P(y \mid x_i, x_o)}{P(y \mid x_o)} = \mathbb{E}_{p(x_i|x_o)} D_{\mathrm{KL}}[P(y \mid x_i, x_o)\|P(y \mid x_o)], \tag{8}$$

where the KL divergence between two discrete distributions can be analytically computed. Note $x_i$ is sampled from $p(x_i \mid x_o)$ as before. We again use Monte Carlo estimation for the expectation.

Although the utility can be estimated accurately by equation 6 and equation 8, it involves some overhead especially for long episodes, since we need to calculate them for each candidate feature at each acquisition step. Moreover, each Monte Carlo estimation may require multiple samples. To reduce the computation overhead, we utilize equation 4 and estimate the entropy terms with Gaussian approximations. That is, we approximate $p(x_i \mid x_o)$ and $p(x_i \mid y, x_o)$ as Gaussian distributions and entropies reduce to a function of the variance. We use sample variance as an approximation. We found that this Gaussian entropy approximation performs comparably while being much faster.

### 2.1.2 SURROGATE MODEL FOR AIR

For unsupervised tasks, our goal is to represent the full set of features with an actively selected subset. Since the target is also $x$, we modify our surrogate model to capture arbitrary conditional distributions $p(x_u \mid x_o)$, which again can be learned using an ACFlow model. Note that by plugging in $y = x$ to equation 4, the utility simplifies to the entropy of unobserved features, which is essentially their uncertainties.

$$\mathcal{U}_i = H(x_i \mid x_o) - \mathbb{E}_{p(x|x_o)}H(x_i \mid x, x_o) = H(x_i \mid x_o) - \mathbb{E}_{p(x_u|x_o)}H(x_i \mid x) = H(x_i \mid x_o). \tag{9}$$

The last equality is due to the fact that $H(x_i \mid x) = 0$. We again use a Gaussian approximation to estimate the entropy. Therefore, the side information for AIR only contains imputed values and their variances of the unobserved features. Similar to the supervised case, we leverage the surrogate

model to provide the intermediate rewards. Instead of using the information gain in equation 3, we use the reduction of negative log likelihood per dimension, i.e.,

$$r_m(s, i) = \frac{-\log p(x_u \mid x_o)}{|u|} - \gamma \frac{-\log p(x_{u \setminus i} \mid x_o, x_i)}{|u| - 1}, \tag{10}$$

since equation 3 involves estimating the entropy for potentially high dimensional distributions, which itself is an open problem (Kybic, 2007). We show in appendix A that the optimal policy is invariant under this form of intermediate rewards. The final reward $-\mathcal{L}(\hat{x}, x)$ is calculated as the negative MSE of unobserved features $-\mathcal{L}(\hat{x}, x) = -\|\hat{x}_u - x_u\|_2^2$.

## 2.2 AFA FOR TIME SERIES

In this section, we apply our GSMRL framework on time series data. For example, consider a scenario where sensors are deployed in the field with very limited power. We would like the sensors to decide when to put themselves online to collect data. The goal is to make as few acquisitions as possible while still making an accurate prediction. In contrast to ordinary vector data, the acquired features must follow a chronological order, i.e., the newly acquired feature $i$ must occur after all elements of $o$ (since we may not go back in time to turn on sensors). In this case, it is detrimental to acquire a feature that occurs very late in an early acquisition step, since we will lose the opportunity to observe features ahead of it. The chronological constraint in action space removes all the features behind the acquired features from the candidate set. For example, after acquiring feature $t$, features $\{1, \ldots, t\}$ are no longer considered as candidates for the next acquisition.

## 2.3 IMPLEMENTATION

We implement our GSMRL framework using the Proximal Policy Optimization (PPO) algorithm (Schulman et al., 2017). The policy network takes in a set of observed features and a set of auxiliary information from the surrogate model, extracts a set embedding from them using the set transformer (Lee et al., 2019), and outputs the actions. The critic network that estimates the value function shares the same set embedding as the policy network. To help learn useful representations, we also use the same set embedding as inputs for the prediction model $f_\theta$.

To reflect the fact that acquiring the same feature repeatedly is redundant, we manually remove those acquired features from the candidate set. For time-series data, the acquired features must follow the chronological order since we cannot go back in time to acquire another feature, therefore we need to remove all the features behind the acquired features from the candidate set. Similar spatial constraints can also be applied for spatial data. To satisfy those constraints, we manually set the probabilities of the invalid actions to zeros.

## 3 RELATED WORKS

**Active Learning**  Active learning (Fu et al., 2013; Konyushkova et al., 2017; Yoo & Kweon, 2019) is a related approach in ML to gather more information when a learner can query an oracle for the true label, $y$, of a complete feature vector $x \in \mathbb{R}^d$ to build a better estimator. However, our methods consider queries to the environment for the feature value corresponding to an unobserved feature dimension, $i$, in order to provide a better prediction on the current instance. Thus, while the active learning paradigm queries an oracle *during training* to build a classifier with complete features, our paradigm queries the environment *at evaluation* to obtain missing features of a current instance to help its current assessment.

**Feature Selection**  Feature selection (Miao & Niu, 2016; Li et al., 2017; Cai et al., 2018), ascertains a static subset of important features to eliminate redundancies, which can help reduce computation and improve generalization. Feature selection methods choose a *fixed* subset of features $s \subseteq \{1, \ldots, d\}$, and always predict $y$ using this same subset of feature values, $x_s$. In contrast, our model considers a *dynamic* subset of features that is sequentially chosen and personalized on an instance-by-instance basis to increase useful information. It is worth noting that our method may be applied after an initial feature selection preprocessing step to reduce the search space.

**Active Feature Acquisition**  Instead of predicting the target passively using collected features, previous works have explored actively acquiring features in the cost-sensitive setting. Ling et al. (2004), Chai et al. (2004) and Nan et al. (2014) propose decision tree, naive Bayes and maximum margin based classifiers respectively to jointly minimize the misclassification cost and feature acquisition

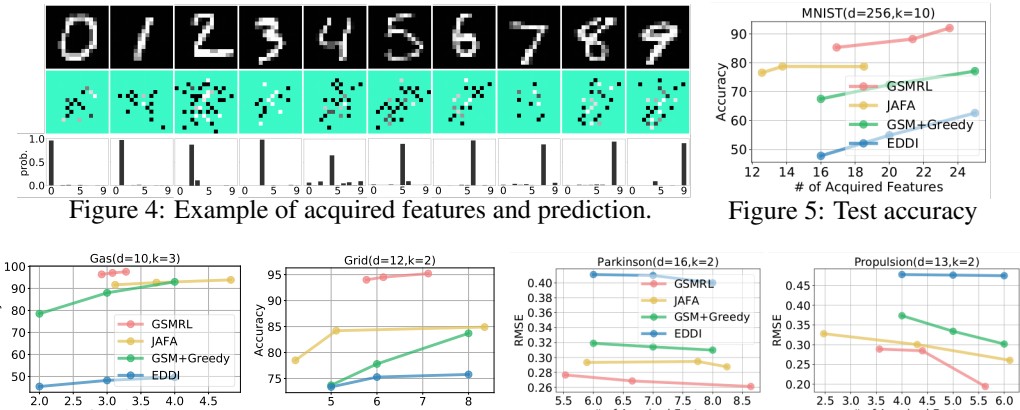

Figure 4: Example of acquired features and prediction.

Figure 5: Test accuracy

Figure 6: Test accuracy on UCI datasets.

Figure 7: Test RMSE on UCI datasets.

cost. Ma et al. (2018) and Gong et al. (2019) acquire features greedily using mutual information as the estimated utility. Zubek et al. (2004) formulate the AFA problem as a MDP and fit a transition model using complete data, then they use the AO* heuristic search algorithm to find an optimal policy. Rückstieß et al. (2011) formulate the problem as partially observable MDP and solve it using Fitted Q-Iteration. He et al. (2012) and He et al. (2016) instead employ the imitation learning approach guided by a greedy reference policy. Shim et al. (2018) utilize Deep Q-Learning and jointly learn a policy and a classifier. The classifier is treated as an environment that calculates the classification loss as the reward. Similar to ours, ODIN (Zannone et al., 2019) presents an approach to combine the model-based and model-free RL for the active feaure acquisition problem. They train a Partial VAE (Ma et al., 2018) as the dynamics model to generate synthetic data and learn a policy and a prediction model using those generated data. In our method, however, a surrogate model, which estimates both the state transitions and the prediction in a unified model, is utilized to provide intermediate rewards and auxiliary information.

**Active Perception**  Active perception is a relevant field in robotics where a robot with a mounted camera is planning by selecting the best next view (Bajcsy, 1988; Aloimonos et al., 1988). Reinforcement learning based active perception models (Cheng et al., 2018; Jayaraman & Grauman, 2018) have recently been proposed to select the next view. In future work, we will explore applying our method for active vision.

**Model-based and Model-free RL**  Reinforcement learning can be roughly grouped into model-based methods and model-free methods depending on whether they use a transition model (Li, 2017). Model-based methods are more data efficient but could suffer from significant bias if the dynamics are misspecified. On the contrary, model-free methods can handle arbitrary dynamic system but typically requires substantially more data samples. There have been works that combine model-free and model-based methods to compensate with each other. The usage of the model includes generating synthetic samples to learn a policy (Gu et al., 2016), back-propagating the reward to the policy along a trajectory (Heess et al., 2015), and planning (Chebotar et al., 2017; Pong et al., 2018). In this work, we rely on the model to provide intermediate rewards and side information.

## 4 EXPERIMENTS

In this section, we evaluate our method on several benchmark environments built upon the UCI repository (Dua & Graff, 2017) and MNIST dataset (LeCun, 1998). We compare our method to another RL based approach, JAFA (Shim et al., 2018), which jointly trains an agent and a classifier. We also compare to a greedy policy EDDI (Ma et al., 2018) that estimates the utility for each candidate feature using a VAE based model and selects one feature with the highest utility at each acquisition step. As a baseline, we also acquire features greedily using our surrogate model that estimates the utility following equation 6, equation 8 and equation 9. We use a fixed cost for each feature and report multiple results with different $\alpha$ in equation 1 to control the trade-off between task performance and acquisition cost. We cross validate the best architecture and hyperparameters for baselines. Architectural details, hyperparameters and sensitivity analysis are provided in Appendix.

**Classification**  We first perform classification on the MNIST dataset. We downsample the original images to $16 \times 16$ to reduce the action space. Fig. 4 illustrates several examples of the acquired

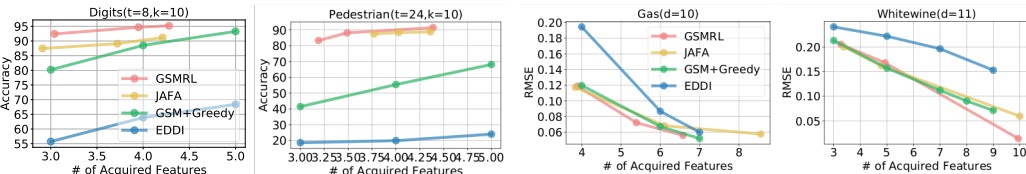

Figure 8: Classification on time series.    Figure 9: RMSE for unsupervised tasks.

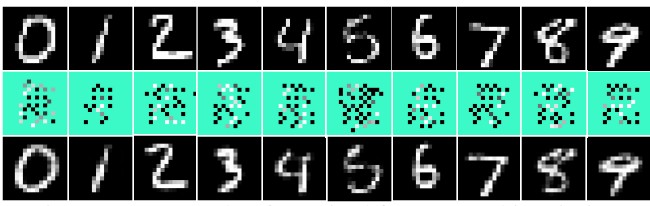

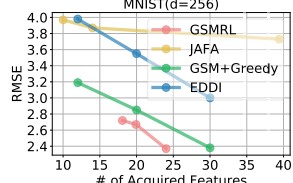

Figure 10: Example of acquired features and inpaintings.    Figure 11: RMSE of $x_u$

features and their prediction probability for different images. We can see that our model acquires a different subset of features for different images. Notice the checkerboard patterns of the acquired features, which indicates our model is able to exploit the spatial correlation of the data. Fig. 1 shows the acquisition process and the prediction probability along the acquisition. We can see the prediction become certain after acquiring only a small subset of features. The test accuracy in Fig. 5 demonstrates the superiority of our method over other baselines. It typically achieves higher accuracy with a lower acquisition cost. It is worth noting that our surrogate model with a greedy acquisition policy outperforms EDDI. We believe the improvement is due to the better distribution modeling ability of ACFlow so that the utility and the prediction can be more accurately estimated. We also perform classification using several UCI datasets. The test accuracy is presented in Fig. 6. Again, our method outperforms baselines under the same acquisition budget.

**Regression**    We also conduct experiments for regression tasks using several UCI datasets. We report the root mean squared error (RMSE) of the target variable in Fig. 7. Similar to the classification task, our model outperforms baselines with a lower acquisition cost.

**Time Series**    To evaluate the performance with constraints in action space, we classify over time series data where the acquired features must follow chronological ordering. The datasets are from the UEA & UCR time series classification repository (Bagnall et al., 2017). For GSMRL and JAFA, we clip the probability of invalid actions to zero; for the greedy method, we use a prior to bias the selection towards earlier time points. Please refer to appendix B.3 for details. Fig. 8 shows the accuracy with different numbers of acquired features. Our method achieves high accuracy by collecting a small subset of the features.

**Unsupervised**    Next, we evaluate our method on unsupervised tasks where features are actively acquired to impute the unobserved features. We use negative MSE as the reward for GSMRL and JAFA. The greedy policy calculates the utility following equation 9. For low dimensional UCI datasets, our method is comparable to baselines as shown in Fig. 9; but for the high dimensional case, as shown in Fig. 11, our method is doing better. Note JAFA is worse than the greedy policy for MNIST. We found it hard to train the policy and the reconstruction model jointly without the help of the surrogate model in this case. See Fig. 2 for an example of the acquisition process.

## 5    ABLATIONS

In this section, we conduct a series of ablation studies to explore the ability of our GSMRL model.

**Surrogate Models**    Our method relies on the surrogate model to provide intermediate rewards and auxiliary information. To better understand the contributions each component does to the overall framework, we conduct ablation studies using the MNIST dataset. We gradually drop one component from the full model and report the results in Fig. 12. The 'Full Model' uses both intermediate rewards

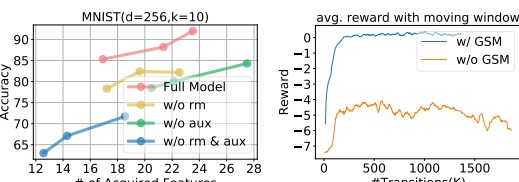

Figure 12: Ablations    Figure 13: Rewards

and auxiliary information. We then drop the intermediate rewards and denote it as 'w/o rm'. The model without auxiliary information is denoted as 'w/o aux'. We further drop both components and denote it as 'w/o rm & aux'. From Fig. 12, we see these two components contribute significantly to the final results. We also compare models with and without the surrogate model. For models without a surrogate model, we train a classifier jointly with the agent as in JAFA. We plot the smoothed rewards using moving window average during training in Fig. 13. We can see the agent with a surrogate model not only produces higher and smoother rewards but also converges faster.

**Dynamic vs. Static Acquisition** Our GSMRL acquires features following a dynamic order where it eventually acquires different features for different instances. A dynamic acquisition policy should perform better than a static one (i.e., the same set of features are acquired for each instance), since the dynamic policy allows the acquisition to be specifically adapted to the corresponding instance. To verify this is actually the case, we compare the dynamic and static acquisition under a greedy policy for MNIST classification. Similar to the dynamic greedy policy, the static acquisition policy acquires the feature with maximum utility at each step, but the utility is averaged over the whole testing set, therefore the same acquisition order is adopted for the whole testing set. Figure 14 shows the classification accuracy for both EDDI and GSM under a greedy acquisition policy. We can see the dynamic policy is always better than the corresponding static one. It is also worth noting that our GSM with a static acquisition can already outperform the EDDI with a dynamic one.

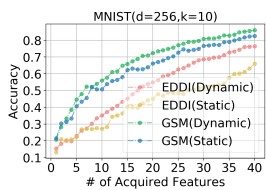

Figure 14: Comparison between dynamic and static acquisition under greedy policies.

**Greedy vs. Non-greedy Acquisition** Our GSMRL will terminate the acquisition process if the agent deems the current acquisition achieves the optimal trade-off between the prediction performance and the acquisition cost. To evaluate how much the termination action affects the performance and to directly compare with the greedy policies under the same acquisition budget, we conduct an ablation study that removes the termination action and gives the agent a hard acquisition budget (i.e., the number of features it acquires). Figure 15 shows the accuracy of MNIST classification with different acquisition budget. We can see our GSMRL consistently outperforms the greedy policy under all different budget, which verifies the benefits of using the non-greedy policy.

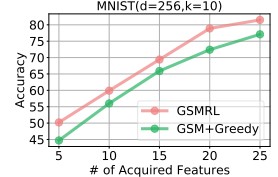

Figure 15: Acquisition with fixed budget.

**Small vs. Large Action Space** For the sake of comparison, we employ a downsampled version of MNIST in the experiment section. Here, we show that our GSMRL model can be easily scaled up to a large action space. We conduct experiments using the original MNIST of size $28 \times 28$. We observe that JAFA has difficulty in scaling to this large action space, the agent either acquires no feature or acquires all features. The greedy approaches are also hard to scale, since at each acquisition step, the greedy policy will need to compute the utilities for every unobserved features, which incurs a total $O(d^2)$ complexity. In contrast, our GSMRL only has $O(d)$ complexity. Furthermore, with the help of

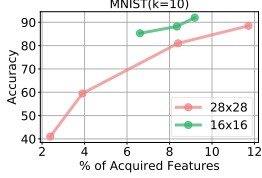

Figure 16: Acquisition with large action space.

the surrogate model, our GSMRL is pretty stable during training and converges to the optimal policy quickly. Fig. 16 shows the accuracy with a certain percent of features acquired. The task is definitely harder for large action space as can be seen from the drop in performance when the agent acquires the same percentage of features for both small and large action space, but our GSMRL still achieves high accuracy by only acquiring a small portion of features.

## 6 CONCLUSION

In this work, we formulate the active feature acquisition problem as an MDP and propose to combine model-based and model-free RL into a holistic framework to resolve the problem. We leverage a generative surrogate model to capture the state transitions across arbitrary feature subsets. Our surrogate model also provides auxiliary information and intermediate rewards to assist the agent. We evaluate our framework on both supervised and unsupervised AFA problems and achieve state-of-the-art performance on both tasks. In future work, we will extend our framework to actively acquire features in spatial-temporal setting, where features are indexed with continuous positions.

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
