# OpenReview forum: "Active Feature Acquisition with Generative Surrogate Models"
_ICLR.cc/2021/Conference — Reject_

### Official Review · AnonReviewer1 · 2020-10-23
**Well-written paper; very incremental contribution**

**Rating:** 6
**Confidence:** 4

**Review:**

This work present a RL and ACFlow based solution for feature wise information acquisition.

Pros:
1. Clarity: The paper is very well presented. Very clear presentation of the problem formulation  and methods.
2. significance: the paper is addressing a very important question in real-world applications where information are associated with cost (which was clearly motivated)
3. significance: the method obtained improved results comparing to current state-of-the-arts methods.

Cons:
originality: very incremental contribution:
* The RL formulation of the problem has been presented in JAFA (Shim et.al 2018). Comparing to Shim et.al, the contribution seems mainly the additional of the surrogate model to do probabilistic imputation/prediction and use the accumulated reward instead of the final ine
* All the claimed contribution comparing to Shim et.al has been done in
ODIN: Optimal Discovery of High-value INformation Using Model-based Deep Reinforcement Learning  https://realworld-sdm.github.io/paper/21.pdf
In ODIN, the surrogate model PVAE was used and it also used PPO.
Thus, the only main difference comparing this paper to ODIN is the choice of surrogate model
* Lastly, the surrogate model is not new. It has been used in paper such as Dynamic Feature Acquisition with Arbitrary Conditional Flows
 Thus, the contribution is really incremental.


 Other small questions:
1. How to read the points on your method in e.g. Figure 5,6,7.  As your method learns the stopping states. Isn't you just get an mean and std for the steps that your method stops.
2. How did you choose the stoping points for GSM-greedy. Did you set some heuristic for greedy method to stop or it is just manual chosen points to fit the plot.
3. I would wish to see how much Non-Greedy contribute and how much stopping action contributed separately. You can remove the stopping action and let it run to the end and compare GSM-RL and GSM-greedy. In that case, it shows how myopic is not so optimal.

---

> ### Author Response · Authors · 2020-11-18
> **Response for AnonReviewer1**
>
> Thanks for your time and the helpful comments.
>
> The MDP formulation for the active feature acquisition problem is first proposed in Zubek et al. (2004), and the RL-based approach has already been explored in Ruckstieß et al. (2011). But we would like to stress the importance and the novelty of leveraging a surrogate dynamics model. As can be seen from the experiments and the ablations, the intermediate rewards and auxiliary information provided by the surrogate model improve the performance by a large margin. In addition, the surrogate model helps to stabilize the training and makes the RL agent converge much faster, which we believe is an important factor for this type of model to be of practical use. In contrast, we observe that JAFA tends to converge to a suboptimal policy where no feature or all features are acquired especially when the action space is high (e.g. for images).
>
> Thanks for pointing us to a relevant paper, but we identify several critical differences between our GSMRL and ODIN. First, ODIN uses a model to generate synthetic data and trains a policy subsequently using those generated data. This is a commonly used way of combining model-based and model-free RL. However, our method utilizes the model to provide auxiliary information about the underlying dynamics, that is the possible values and uncertainties of all the unobserved features. In addition, our method utilizes the model to provide intermediate rewards, which reduces the sparsity of the training rewards and simplifies the credit assignment problem. To the best of our knowledge, our GSMRL presents a novel way of combining model-based and model-free RL, which could be of use for a larger community. Second, ODIN trains two separate networks for the surrogate model and the prediction model respectively, while we unify these two models using Bayes’s rule. Third, for the first time, we scale the AFA problem up to a large action space and image data, while all the previous methods including ODIN only deal with a small action space and vector data. Lastly, in addition to the supervised tasks where features are acquired to predict a target variable, we propose an unsupervised task where features are acquired to impute the unobserved features. We believe this task could have broader applications, like image compression, which we will explore in future works. We also explore the time series acquisition problem, which, to the best of our knowledge, has not been explored by previous methods.
>
> “Dynamic Feature Acquisition with Arbitrary Conditional Flows” is an unpublished work and we use it as one of the baselines (GSM+Greedy). DFA is a greedy method and our method outperforms it on all the datasets. Besides, DFA is computationally expensive especially for large action space since at each acquisition step the model needs to evaluate the utilities for all the remaining unobserved features.
>
> Answers to specific questions:
> 1. According to Eq1, the model will behave differently depending on the relative weights $\alpha$ we give to acquisition cost. Typically, smaller weights will allow the model to acquire more features and make a better prediction. We conduct experiments with three different $\alpha$ for all the datasets to illustrate this trade-off. Each point in the result plots corresponds to a different $\alpha$.
> 2. The points for greedy methods (EDDI and GSM-Greedy) are manually chosen to better visualize the results.
> 3. We add an ablation study that removes the termination action and gives the agent a hard budget for acquisition so that we can directly compare our GSMRL with the greedy policy under the same budget. From Fig.15, we can see the RL based policy outperforms the greedy one under all budgets.

---

### Official Review · AnonReviewer4 · 2020-10-27
**a reasonable solution to active feature acquisition, while the advantages to existing approaches are not clear and the novelty is not enough**

**Rating:** 4
**Confidence:** 4

**Review:**

#### Paper summary:

In this work, a reinforcement learning (RL) approach is proposed to solve the active feature acquisition (AFA) problem (as well as the active instance recognition problem). Comparing to existing RL approaches for AFA, the main difference of the proposed approach is to introduce a generative model (utilizing the existing ACFlow model) to learn the transition function, in order to provide additional rewards and auxiliary information. The proposed approach is evaluated on MNIST and UCI datasets, which can outperform two existing baselines.

#### Advantages:

- I think the paper is well-written. The proposed approach is clearly explained. In my view, the overall workflow of the algorithm is clear. It is a reasonable approach to solve the AFA problem.

- The idea of learning a generative model is interesting. It can be viewed as a good realization of the general idea of active RL: try to identify uncertain states for querying.

#### Disadvantages:

- The advantages of existing approaches are not clearly explained. There are many existing works on AFA as cited in the paper, while they are just listed instead of explaining the relationships to the current work clearly. What is the main challenge for AFA? What are the main drawbacks of the existing approaches? Why RL is necessary? Adding more explanations on these points can be very beneficial.

- From the technical perspective, the proposed approach builds upon existing RL and generative learning algorithms, thus the technical novelty is limited.

- The experiments are not sufficient for the following perspectives:

    - Datasets and comparison methods are limited. What are the reasons to select these specific datasets and comparison methods? In my view, adding more could be more persuasive to show the effectiveness of the proposed algorithm.
    - No results are shown to show what the generative model learned in these experiments. This is crucial to understand what help can the generative model provides in learning.
    - There are no datasets for real AFA tasks included (e.g. the medical treatment task introduced in section 1). This is very important for understanding the true effective tasks for the proposed approach.

- Citations and the discussions on the below AFA works are missing:
   - TEFE: A Time-Efficient Approach to Feature Extraction. https://ieeexplore.ieee.org/document/4781137
   - Unsupervised Sequential Sensor Acquisition. http://proceedings.mlr.press/v54/hanawal17a.html

#### Overall evaluation:

I think the paper proposes a reasonable solution to the AFA problem, while the contributions and novelty are somehow below the standard of ICLR.

#### Minor:
$y'$ in equation 5 might be $y$.

---

> ### Author Response · Authors · 2020-11-18
> **Response for AnonReviewer4**
>
> Thanks for your comments.
>
> Most of the cited works utilize a fundamentally different approach for the AFA problem, like the greedy approach and imitation learning. The greedy method is computationally expensive with a $O(d^2) complexity. The imitation approach requires an oracle policy, which previous methods typically use a greedy one. The most similar method to ours is Shim et al. (2018), in which they utilize a deep Q-learning to optimize the MDP. Our method is different in that we introduce a surrogate model that provides auxiliary information and intermediate rewards. The AFA problem is to acquire features that are unknown prior to us, therefore the biggest challenge is to infer how informative each feature could be for the downstream task before actually acquiring it. Without a surrogate model, the agent would be completely in charge of inferring the informativeness of each unobserved feature based on a sparse reward. However, our surrogate model directly assesses the informativeness using the information gain and provides it as intermediate rewards. In addition, our surrogate model provides some useful auxiliary information, such as the imputed values of unobserved features and an estimate of the expected information gain for each unobserved feature. The auxiliary information can make a big difference for the AFA problem. As can be seen from our reported results that a greedy policy that selects the feature with the highest expected information gain can already achieve competitive performance. Equipped with the auxiliary information, our agent can implicitly plan ahead and refine the greedy policy, therefore our GSMRL achieves better results compared to the greedy one on all the datasets. Furthermore, the surrogate model stabilizes the training, makes the RL agent converge faster, and makes it possible to scale up to a larger action space, which resolves the main drawback of JAFA.
>
> The PPO algorithm and the ACFlow model are indeed existing techniques, but it is not immediately clear about how to combine them. We are, to the best of our knowledge, the first one to utilize a surrogate model to provide auxiliary information and intermediate rewards for solving this challenging AFA problem. As we mentioned above, these two algorithms compensate for each other to obtain a better solution, which neither of them can achieve alone. Our GSMRL represents a new way of combining model-based and model-free RL, which can be of use for a larger community. Furthermore, the ACFlow is not entirely an existing model. We extend it based on Bayes rule to deal with both the generative and the discriminative side of the problem. Our extended ACFlow can handle  both continuous and discrete target variables.
>
> Our experiments are the largest scale study to date for the AFA problem. Previous works are all dealing with smaller datasets (both in terms of the number of features and the number of instances). We also compare to the current state-of-the-art models, while previous works only compare to simple baselines, such as random acquisition order.
>
> The imputed images with several different observed features are shown in Fig.2, Fig.10, Fig.D.3 and Fig.D.4. It shows that the surrogate model learns a reasonably good imputation model. When the observed feature set is small, it captures the ambiguity and imputation tends to be blurry. When the agent acquires more features, the imputation gets better quickly.
>
> We agree that results on medical datasets would be very interesting to see. However, our focus in this work is to develop algorithms that are as general as possible. We would explore some medical applications in future works.

---

### Official Review · AnonReviewer2 · 2020-10-29
**Feature acquisition method that relies on RL with auxiliary rewards**

**Rating:** 5
**Confidence:** 4

**Review:**

This paper operates in the setting where there is a possibility to adaptively acquire features for the prediction on each datapoint. The authors study classification, regression, time series and feature completion problems. The proposed solution relies on RL and introduces additional hand crafted rewards and features.

Strong points:
- The paper tries to solve an important problem that deserves additional attention from the researchers.
- The experimental results are promising and the proposed method outperforms the baselines in several settings. The experimental results include several datasets and types of the problems.
- The approach has shown itself being quite generic in a sense that it is applicable to different prediction problems, such as classification, regression, time series and feature completion.

Weak points:

- The datasets for the experiments are rather simple and have only a handful of dimensions (except for MNIST, but even MNIST is downsampled to 16x16). It makes me wonder how the proposed method would scale to high-dimensional realistic datasets.

- The approach is rather heuristic and relies heavily (as shown in the ablations) on the hand-crafted features for the state representation and engineered rewards. Another heuristic part is the prediction model f_\theta that is sometimes used to make predictions and sometimes not.

- In the experiments with MNIST the features are the different pixels in the images. However, this is a very special type of the features and many methods that rely on the inductive bias about images could be used in addition. As it is the most complex experiment in this paper, I think the related works that are specific for images could be mentioned. For example, see [1] (but there are many works in this problem setting).

- The auxiliary loss used in the paper seems to be quite intuitive and helps to eliminate the problem of sparse rewards. However, I am wondering if this auxiliary reward actually modifies the optimal solution to the original MDP [2].

- The experimental section is quite short and does not provide empirical evidence towards understanding of the benefits of the proposed method. For example, the authors state that the approach learns a "non-greedy policy". It would be informative to see how different the queried features are from the greedy selection. Then, the authors mention the superiority of the adaptive feature acquisition over selecting a fixed set of features for all datapoints. While the argument seems to be very logical, it would be nice to see the empirical confirmation of this and how big the gain of the proposed adaptive approach is.

- While the paper is reasonably well written at the sentence level, the structure of the paper is a bit hard to follow. For example, the abstract and introduction go into many details about the method, which are hard to understand at that point. Then, it feels that the methods section repeats a lot of things that are already described (at the same level of details). The experimental section is on the contrary very short (for example, it does not even mention what the dataset for time series experiments is and refers directly to the appendix).

I am leaning towards the rejection of this paper. While I appreciate the challenges of the studied problem and the proposed solution based on RL, I think the paper would benefit from 1) better motivation of the proposed techniques, 2) more experimental analysis to support the claims of the paper, 3) experimental evaluation on more complex datasets, and 4) some restructuring to improve the reader's understanding.

Questions:
- What is the computational complexity of this method and how would it scale to more realistic datasets with many features?
- Could the authors elaborate on what effect the auxiliary loss has on the original MDP?
- Could the authors elaborate on the prediction function f_\theta and explain when it is used and when not and why?
- I am not entirely convinced that using the proposed "surrogate model" shares a lot in common with model-based RL methods. For example, "surrogate model" does not encode much about the transition dynamics in the environment. Could the authors elaborate more on this?

Additional comments:
- I am not completely convinced by the argument with "limited power" of the sensors in time series predictions, especially in the context of the potential computational cost of the method.
- Scales in figures vary, maybe the most informative way of selecting the scale would be to put the upper/lower bound that a method with all features can achieve?
- I didn't understand how varying \alpha is reflected in the plots.

[1] Learning to Look Around: Intelligently Exploring Unseen Environments for Unknown Tasks, Dinesh Jayaraman, Kristen Grauman. CVPR, 2018.
[2] Policy invariance under reward transformations: Theory and application to reward shaping, Andrew Ng, Daishi Harada, Stuart Russell. ICML, 1999.

--- After reading the authors' response ---

The authors' response addressed some of my concerns. However, I could see that some of the concerns regarding novelty, relation to the prior work and experimental results are shared among several reviewers. Thus, I keep my original score and I believe that the revised manuscript would benefit from another round of reviewing.

---

> ### Author Response · Authors · 2020-11-18
> **Response for AnonReviewer2**
>
> Thanks for your time and the helpful review.
>
> We would like to note that our experiments consider the largest scale study to date. Previous work only considered smaller datasets. In addition, our method can be scaled up to a larger action space (see Fig.16). The reason why we use a downsampled version is that the baselines are hard to scale up. The greedy methods are computationally demanding since at each acquisition step the model needs to evaluate the utilities for all the unobserved features, which incurs a total $O(d^2)$ complexity. JAFA is unstable even on this downsampled dataset, and it is even harder to train with higher-dimensional action space. In contrast, our method only incurs a $O(d)$ complexity and is pretty stable during training.
>
> Our surrogate model does capture the state transitions. It learns the arbitrary conditional distributions $p(x_u \mid x_o)$, where $x_o$ represents the current observed features (states) and $x_u$ represents arbitrary unobserved features. The surrogate model captures the state transitions $s_t \xrightarrow{a} s_{t+1}$ since the features acquired by action $a$ is also part of $x_u$. Therefore, the auxiliary information is not just some arbitrary hand-crafted state representation. It provides the imputed values and uncertainties of the unobserved features. Furthermore, the surrogate model provides an estimate of the expected information gain for each unobserved feature, which essentially estimates how informative each unobserved feature is for the downstream task. The intermediate rewards are also more than just a heuristic. It represents the information gain of the current acquired feature, which is a theoretically grounded metric to evaluate the informativeness of a feature. As a matter of fact, armed with the ability to impute unobserved features and evaluate their expected informativeness, we can build a greedy policy (GSM+Greedy) where the most informative feature is selected at each acquisition step. Our greedy policy is already very competitive and outperforms EDDI on all the datasets, and our GSMRL further outperforms the GSM+Greedy by a large margin.
>
> We stated the prediction model as optional since the surrogate model can already provide a prediction. But the surrogate model is trained to deal with arbitrary observed features and is not specifically tuned to the observed features the agent may encounter, therefore we think an additional prediction model may help. We always use a prediction model during experiments, and we found the prediction model is giving better predictions than the surrogate model for all the UCI datasets but is worse for MNIST. It is probably because of a very simple architecture we used for the prediction model. However, even for MNIST, we found the prediction model does not hurt the learning process. We believe the prediction model is acting like an auxiliary task, which helps the policy network to extract better representations from the state.
>
> Thanks for pointing us to the theoretical work about policy invariance. Based on their theorem, we can actually show that our intermediate rewards will not change the optimal policy. Please refer to the revised appendix for more details. Note that we introduce a discount factor to our intermediate rewards in Eq3 and 10 and rerun our models. Since the discount factor is set to 0.99, the results do not change much compared to what we originally reported in the paper.
>
> We add an ablation study to compare the dynamic acquisition and the static acquisition (a fixed acquisition order for all instances). The results show that the dynamic policy performs consistently better than the static one (see Fig.14). We also add an ablation study where a fixed acquisition budget is specified for the agent so that we can directly compare the greedy and non-greedy policy under the same budget. Results in Fig.15 suggests the non-greedy policy consistently outperforms the greedy one for all budgets. Furthermore, we show several examples of the acquisition process in the appendix to compare the greedy and non-greedy policy. Fig.D.1 and Fig.D.2 show the acquisition process for the AFA task. Fig.D.3 and Fig.D.4 show the acquisition process for the AIR task. From those results, we can see that the non-greedy policy eliminates the prediction uncertainty much faster than the greedy one.
>
> Answers to specific questions:
> 1. During the testing phase, we only need to run the model in a forward direction, which is not insanely expensive. Besides, we believe there are scenarios that sensing costs much more energy (or money, risk) than computation.
> 2. According to Eq1, the model will behave differently depending on the relative weights $\alpha$ we give to acquisition cost. Typically, smaller weights will allow the model to acquire more features and make a better prediction. We conduct experiments with three different $\alpha$ for all the datasets to illustrate this trade-off. Each point corresponds to a different $\alpha$.

---

### Official Review · AnonReviewer3 · 2020-10-29
**Active Feature Acquisition with Generative Surrogate Models**

**Rating:** 7
**Confidence:** 4

**Review:**

This paper studies the problem of active feature acquisition (AFA). The authors formulate AFA as a Markov decision process (MDF) and use reinforcement learning to resolve it. In order to overcome the sparse reward and complicated action space in this situation, the authors combine a generative surrogate model into their framework (GSMRL) to provide more feedback to the agent. Additionally, the authors adapt GSMRL into the supervised, unsupervised task (AIR) domain and introduce the corresponding dealing process in detail. Finally, the authors conduct lots of experiments to validate the superiority of GSMRL and the necessity of each component in GSMRL.

The research direction of this paper is important and interesting. In real scenarios, there are many situations in which people cannot observe all features before making a decision or constructing a model. The framework of this paper is flexible and adaptable to these situations. Therefore, this paper is useful for solving real problems.

Pros:
1.	The presentation of this paper is good. It is easy for readers to follow the logic and the main idea of the paper.
2.	The authors provide many experimental details. Readers can reproduce the experimental results based on them easily.
3.	The authors conduct a lot of experiments from different perspectives to validate the effectiveness of their work.
Cons:
1.	The authors should modify their symbol system. For example, when they introduce the surrogate model. They need to construct a model to produce conditional distribution. At there, X_o and X_u are a subset of the features. I recommend making the font of X_u and X_o bold. I think this modification will make people get the idea of the formula easier.
2.	AFA For Time Series in Section 2 Methods is too simplified. But from the chronological of this paper, this part is paralleled with AFA and AIR with GSMRL. I recommend the authors to consider how to organize the component of this part to make the paper more logical.

In summary, this paper conducts interesting and vital research. GSMRL is flexible and effective for many tasks. The authors provide an organized and qualified introduction for their framework. Therefore, I recommend accepting this paper.

---

> ### Author Response · Authors · 2020-11-18
> **Response for AnonReviewer3**
>
> Thanks for your time and affirmation. We will consider your suggestion about the symbolic system. If space permits, we will move some details about the time series acquisition from the appendix to the main part.

---

### Author Response · Authors · 2020-11-18
**General Response**

We appreciate all reviewers for their helpful comments.

We would like to stress some of the main contributions of our GSMRL model.
- First, we propose a new way of combining model-based and model-free RL to solve the active feature acquisition problem. Although RL based methods and generative models are already being studied for this problem, it is not immediately clear about how to combine them. Instead of using the generative model to sample synthetic training data like ODIN does, we propose leveraging a generative surrogate model to provide auxiliary information and intermediate rewards. Our method represents a novel way of integrating models into a policy learning process, which could be of use for a larger community. Furthermore, equipped with the surrogate model, our RL agent is stable during training and converges much faster, which is one of the drawbacks of current RL based approaches.
- Second, we consider the largest scale study to date for the active feature acquisition problem. Previous works only consider small datasets (both in terms of the number of features and the number of instances). We instead consider a broad range of datasets with much more instances and much higher dimensionality. In terms of the comparison, previous works compare to naively simple baselines, such as a random acquisition order. In this work, we compare our GSMRL to the state-of-the-art models with both greedy policy and non-greedy RL policy.
- Third, we consider the feature acquisition with constraints in the action space. Chronological and spatial constraints abound much in real-world applications, but they have not been studied yet in the literature.
- Last but not least, we propose a new problem setting where we do not have a specific target variable of interest. Instead, we acquire features to reveal information about the instance itself. That is, we care about the holistic features of the instance, not just those related to the target variable. The active instance recognition (AIR) problem setting is a novel extension of the AFA problem and potentially has many applications, such as data compression.

---

### Decision · Program_Chairs · 2021-01-07
**Final Decision**

**Decision:**

Reject

**Comment:**

The paper studied an interesting and important problem in active learning/information acquisition (AFA), and provided an RL-based active learning scheme for a broad spectrum of AFA tasks, in both supervised (active classification/regression) and unsupervised (feature completion/recovery) domains. The reviewers generally find the paper well presented, and all appreciate the broad applicability of the proposed approach, which leverages reinforcement learning and a generative surrogate model to learn the acquisition/reward function of AFA. However, there are also shared concerns among several reviewers on the novelty and positioning of the proposed approach, as well as on whether the proposed experiments results well demonstrated the significance of the algorithm. Given that this is a purely empirical paper, both aspects are important to be properly addressed in a revision.